Stable hydrogen isotopes record the summering grounds of eastern red bats (Lasiurus borealis)

Pylant Cortney L. 1 2
Nelson David M. 2 dnelson@umces.edu
Keller Stephen R. 2
1 Department of Biology, Frostburg State University , Frostburg, MD , USA
2 Appalachian Laboratory, University of Maryland Center for Environmental Science , Frostburg, MD , USA
Gandini Patricia
Electronic publication date: 2014 Oct 16
Publication date: 2014
Volume: 2
Electronic Location ID: e629
Received 2014 Jul 30; Accepted 2014 Sep 30
Copyright: © 2014 Pylant et al.
Copyright year: 2014
Copyright holder: Pylant et al.
License: This is an open access article distributed under the terms of the Creative Commons Attribution License, which permits unrestricted use, distribution, reproduction and adaptation in any medium and for any purpose provided that it is properly attributed. For attribution, the original author(s), title, publication source (PeerJ) and either DOI or URL of the article must be cited.
License URL: https://creativecommons.org/licenses/by/4.0/

Keywords: Lasiurus borealis, Migration, Eastern red bat, Stable hydrogen isotopes, Wind energy

Funding: The Maryland Department of Natural Resources EPA STAR program FP-91749401-0 The Maryland Department of Natural Resources and EPA STAR program (fellowship FP-91749401-0) provided funding for this research. The funders had no role in study design, data collection and analysis, decision to publish, or preparation of the manuscript.

==============================
Bats face numerous threats associated with global environmental change, including the rapid expansion of wind-energy facilities, emerging infectious disease, and habitat loss. An understanding of the movement and migration patterns of these highly dispersive animals would help reveal how spatially localized the impacts from these threats are likely to be on bat populations, thus aiding in their conservation. Stable hydrogen isotope ratios (δ2H) can be used to infer regions where bats have foraged during the summer molt season, thus allowing an assessment of summering location and distance of movement of bats sampled during other times of year. However, a major impediment to the application of δ2H for inference of bat movements is that the relationship between δ2H of bat hair and precipitation tends to be species specific and is still unknown for some key species of conservation concern. We addressed this issue by using geo-referenced museum specimens to calibrate the relationship between δ2H of hair (δ2Hhair) and long-term δ2H of growing-season precipitation (δ2HGSprecip) at the site of collection for eastern red bats (Lasiurus borealis), one of the main species of bats experiencing large numbers of fatalities at wind-energy facilities in North America. Based on comparison of δ2Hhair and δ2HGSprecip values for males we estimated a period of molt of June 14–August 7. Within this period, male and female red bats exhibited a significant positive relationship between δ2Hhair and δ2HGSprecip. These results establish the relationship between δ2Hhair and δ2HGSprecip for red bats, which is necessary for the use of δ2Hhair to infer the movement and migration patterns of this important species. These results provide a critical resource to conservation biologists working to assess the impacts of environmental change on bat populations.

Introduction

Bats living in temperate zones display a range of strategies for escaping unfavorable winter conditions. Species that survive through winter by hibernating in caves or buildings often have relatively sedentary populations or undertake regional migrations, whereas other species undertake seasonal migrations of hundreds or thousands of kilometers to find suitable winter habitat (Fleming & Eby, 2003; Cryan et al., 2004). However, despite the importance of movement and migration patterns to bat ecology and conservation, such behaviors remain difficult to quantify for these highly dispersive animals.

Understanding patterns of movement and migration is key to the conservation of bats experiencing threats associated with global environmental change, including the rapid worldwide expansion of wind-energy facilities, emerging infectious disease, and habitat loss (Kuvlesky et al., 2007; NRC, 2007; Cryan & Barclay, 2009; Boyles et al., 2011). Such information may aid bat conservation by helping to reveal migration pathways, population connectivity, regional habitat use, and the spatial extent of the impacts from these threats (Webster et al., 2002; O’Shae, Bogan & Ellison, 2003). Unfortunately, existing tracking methods are of limited use for understanding movement and migration of bats that migrate long distances. For example, mark-recapture studies suffer from low recapture rates (Holland & Wikelski, 2009). The use of radio transmitters suffers from small ranges of detection and short life spans of batteries that limit their ability to track bats capable of migrating long distances (Cryan & Diehl, 2009; Taylor et al., 2011; McGuire et al., 2012). Furthermore, geolocation by light is of limited use for nocturnal or crepuscular organisms (Lisovski et al., 2012), such as bats that roost in heavy foliage and are active when there is little to no sunlight. Alternatively, intrinsic markers, such as stable hydrogen isotope ratios (δ2H), overcome these challenges and are a viable method to infer the summering grounds of bats (Cryan et al., 2004; Fraser et al., 2012; Sullivan et al., 2012). The basis of this approach is that continental-scale variation in δ2H of precipitation (Dansgaard, 1964) is incorporated into hair keratin through drinking water and diet (Estep & Dabrowski, 1980; Fogel & Cifuentes, 1993), and this incorporation occurs during summer when temperate bats undergo their annual molt (Quay, 1970; Cryan et al., 2004; Cryan et al., 2012; Fraser, Longstaffe & Fenton, 2013). The use of stable isotopes has several advantages: it can be applied to live and dead bats, only small quantities of hair are required, and geographic origins of summering grounds can be assigned to bats captured outside the period of molt because hair is metabolically inert (Rubenstein & Hobson, 2004).

Application of δ2H to infer the geographic origin of bats requires the prior estimation of isotopic discrimination between δ2H of their tissues and δ2H of precipitation incorporated into their drinking water and diet. To validate this relationship, bats are sampled at their known summering grounds and values of δ2H of hair (δ2Hhair) are compared with values of δ2H of growing-season precipitation (δ2HGSprecip) at the same locations. Prior studies have shown strong positive relationships between δ2Hhair and δ2HGSprecip for hoary bats (Lasiurus cinereus; Cryan et al., 2004; Cryan, Stricker & Wunder, 2014), tri-colored bats (Perimyotis subflavus, Fraser et al., 2012), little brown bats (Myotis lucifugus, Britzke et al., 2009; Sullivan et al., 2012), and others (Britzke et al., 2009; Popa-Lisseanu et al., 2012; Table 1). However, the relationship between δ2Hhair and δ2HGSprecip is often species specific as the result of differences in life history and/or physiology, so the δ2H relationship established for one species is often not transferable to another species (Table 1; Britzke et al., 2009; Hobson et al., 2012).

Table 1 Review of published relationships between δ2Hhair and δ2HGSprecip for North American bats.

Note that the combined regressions from Britzke et al. (2009) include juvenile bats of unknown sex.

Species	Equation	R 2	p-value	Source	
Perimyotis subflavus (tri-colored bat)					
male (n = 29)	δ2Hhair=−0.036∗δDGSprecip2
− (1.79∗δ2HGSprecip)−45.61	0.86	<0.01	(Fraser et al., 2012)	
female (n = 27)	δ2Hhair=−0.034∗δDGSprecip2
− (1.61∗δδ2HGSprecip)−40.38	0.75	<0.01	(Fraser et al., 2012)	
Myotis lucifugus (little brown bat)					
male (n = 12)	δ2Hhair = (0.49∗δ2HGSprecip)−30.90	0.19	0.1527	(Britzke et al., 2009)	
female (n = 54)	δ2Hhair = (0.33∗δ2HGSprecip)−40.41	0.06	0.0492	(Britzke et al., 2009)	
combined (n = 78)	δ2Hhair = (0.52∗δ2HGSprecip)−30.82	0.17	0.0002	(Britzke et al., 2009)	
combined (n = ?)	δ2Hhair = (2.69∗δ2HGSprecip) + 96.93	0.63	<0.001	(Sullivan et al., 2012)	
Myotis septentrionalis
(northern long-eared bat)					
male (n = 10)	δ2Hhair = (0.79∗δ2HGSprecip)−4.73	0.53	0.0088	(Britzke et al., 2009)	
female (n = 16)	δ2Hhair = (1.25∗δ2HGSprecip) + 18.48	0.71	0.0001	(Britzke et al., 2009)	
combined (n = 33)	δ2Hhair = (0.98∗δ2HGSprecip) + 5.48	0.54	<0.001	(Britzke et al., 2009)	
Myotis sodalis (Indiana bat)					
male (n = 12)	δ2Hhair = (0.90∗δ2HGSprecip)−0.59	0.46	0.0115	(Britzke et al., 2009)	
female (n = 39)	δ2Hhair = (0.71∗δ2HGSprecip)−8.17	0.35	0.0001	(Britzke et al., 2009)	
combined (n = 59)	δ2Hhair = (0.83∗δ2HGSprecip)−2.97	0.49	<0.0001	(Britzke et al., 2009)	
Lasiurus cinereus (hoary bat)					
combined (n = 104)	δ2Hhair = (0.79∗δ2HGSprecip)−24.81	0.60	<0.001	(Cryan et al., 2004)	
combined (n = 117)	δ2Hhair = (0.73∗δ2HGSprecip)−42.61	0.55	<0.001	(Cryan, Stricker & Wunder, 2014)	
Lasiurus borealis (eastern red bat)					
male (n = 17)	δ2Hhair = (−0.82∗δ2HGSprecip)−58.80	0.33	0.0482	(Britzke et al., 2009)	
female (n = 36)	δ2Hhair = (1.35∗δ2HGSprecip)−3.60	0.31	0.0003	(Britzke et al., 2009)	
combined (n = 81)	δ2Hhair = (0.48∗δ2HGSprecip)−26.10	0.07	0.0201	(Britzke et al., 2009)	
male (n = 20)	δ2Hhair = (1.48∗δ2HGSprecip) + 13.95	0.69	<0.001	This study	
female (n = 44)	δ2Hhair = (1.75∗δ2HGSprecip) + 18.02	0.29	<0.001	This study	
combined (n = 64)	δ2Hhair = (1.67∗δ2HGSprecip) + 16.84	0.37	<0.001	This study	

The eastern red bat (Lasiurus borealis) is among the bat species experiencing the highest levels of mortality at wind-energy facilities in the eastern United States (Arnett et al., 2008). Red bats are thought to undertake long-distance migrations from their winter grounds along coastal regions of the southeastern United States and the Gulf of Mexico to widely distributed summering grounds located throughout eastern North America (Cryan, 2003). Their northern range limit is southern Canada and their western range limit is the Rocky Mountains (Shump & Shump, 1982; Cryan, 2003). In contrast to most prior studies, Britzke et al. (2009) found a negative relationship between δ2Hhair and δ2HGSprecip for male red bats, but a positive relationship for females. These results suggest that male red bats may have unusual migration patterns and/or isotopic discrimination relative to female red bats and other species, such as hoary bats (Cryan et al., 2004), a sister taxa (Roehrs, Lack & Van Den Bussche, 2010). Since this intraspecific difference is unusual and red bats are a species of conservation concern because of wind-turbine mortality, additional studies are required to assess the relationship between δ2Hhair and δ2HGSprecip for red bats and the applicability of this relationship to assigning geographic origins of migrants. We hypothesized that if δ2Hhair is useful for inferring locations at which red bats summer, then individuals from regions with more negative δ2HGSprecip values should have more negative δ2Hhair values than individuals from regions with more positive δ2HGSprecip values. Further, if male and female red bats exhibit similar migration patterns and patterns of isotopic discrimination, then we expect no difference in their relationships between δ2Hhair and δ2HGSprecip.

Materials and Methods

We searched the Smithsonian Institution National Museum of Natural History’s Division of Mammals Collections database (http://collections.nmnh.si.edu/search/mammals) for red bat specimens that (1) had sufficiently detailed information to be able to geo-reference the location of collection, and (2) were collected during June–August. This period includes the time of year when red bat individuals are most likely to be resident on their summering grounds, as approximated from the estimated period of molt in hoary bats (Cryan et al., 2004) and other bat species (see Fraser, Longstaffe & Fenton, 2013 for a review of published molt dates). The pool of potential specimens was selected to maximize geographic coverage throughout the known distribution of red bats (Fig. 1) and to minimize overrepresentation of samples from similar locations. When available from specimen labels, we recorded the sex of each individual.

Figure 1 Map of collection sites of museum specimens within the known range of L. borealis.

Red diamonds, male; blue circles, female; yellow squares, sex unknown. Solid symbols represent samples of males and females collected between June 14 and August 7 and open symbols represent samples of males and females collected outside of this period, as well as the four bats of unknown sex. Some symbols represent the location at which multiple bats were collected. The dark line represents the approximate geographic range throughout which the red bat occurs (IUCN, 2008).

We removed approximately 1 mg of hair from the axillary region of each specimen to minimize visible damage to the specimens. We cleaned the samples of natural oil and residues using 1:200 Triton X-100 detergent and 100% ethanol. Then, each sample was air dried at ambient temperature, as recommended by Coplen & Qi (2012). To account for exchange of keratin hydrogen with ambient vapor we used a comparative equilibration approach (Wassenaar & Hobson, 2003) in which samples were equilibrated and analyzed alongside international hair standards (USGS42, Tibetan hair, and USGS43, Indian hair; Coplen & Qi, 2012) and an internal keratin standard (porcine hair and skin, Spectrum Chemical product # K3030). Approximately 0.3 mg of cleaned hair from each bat sample, as well as each standard, was weighed into silver capsules and exposed to ambient air for >72 h to allow for equilibration of exchangeable hydrogen. Samples were analyzed for δ2H using a ThermoFisher high temperature conversion/elemental analyzer pyrolysis unit interfaced with a ThermoFisher Delta V+ isotope ratio mass spectrometer at the Central Appalachians Stable Isotope Facility (http://casif.al.umces.edu). Values of δ2H are expressed in parts per mil (‰) using the following equation: δ2H(‰) = [(Rsample/Rstandard−1) × 1,000], where R is the ratio of 2H/1H. δ2H sample data were normalized to the Vienna Standard Mean Ocean Water-Standard Light Antarctic Precipitation (VSMOW-SLAP) scale using a two-point normalization curve with USGS42 and USGS43, whose δ2H values of non-exchangeable hydrogen are −78.5 and −50.3‰, respectively. Most of the δ2Hhair values of the specimens were >−50.3‰, but prior studies suggest that linear extrapolation of normalization relationships for δ2H is appropriate for values within ∼100‰ of the range of the standards used for normalization (Kelly et al., 2009; Wiley et al., 2012). The analytical precision of the internal keratin standard was ±1.9‰.

We used Google Earth to determine the approximate latitude, longitude and elevation of the collection location of each specimen, based on information provided on the specimen labels. Where information was restricted to broader geographic regions (e.g., counties, national parks) we used values for a central point. Latitude, longitude and elevation values were entered in the Online Isotopes in Precipitation Calculator (http://waterisotopes.org; Bowen & Revenaugh, 2003; Bowen, Wassenaar & Hobson, 2005) to determine average δ2H values of precipitation for June–August (i.e., δ2HGSprecip) for each collection site. The small uncertainties associated with our approach for approximating the latitude, longitude and elevation of sample locations had little influence on the δ2H values of precipitation that were calculated for each site because δ2H values of precipitation exhibit greater variation across large than small environmental gradients (e.g., of latitude). Specimen collection years spanned a period from 1900 to 2009. We subset samples by sex for initial analyses to assess potential intersex differences; specimens of unknown sex were excluded from these analyses.

The variance of the difference between δ2Hhair and δ2HGSprecip values should decrease during the period of molt. Therefore, to attempt to more precisely estimate the range of days during which new pelage was presumably synthesized, we empirically evaluated the interval of time during the June–August period for which the standard deviation (created by grouping the Julian days of collection into 5-day intervals) of the difference between individual δ2Hhair and δ2HGSprecip values was minimized. To do this, we calculated the standard deviation of the difference between individual δ2Hhair and δ2HGSprecip values. We determined the presumed period of molt by visually identifying where the standard deviation was the lowest. We included samples collected during the presumed period of molt in subsequent reduced major axis (RMA) regressions. We performed RMA regressions to assess the relationship between δ2Hhair and δ2HGSprecip because of symmetry between the dependent and independent variables (i.e., it is arbitrary which variable is plotted on the X and Y axes, because δ2Hhair is influenced by δ2HGSprecip, but δ2HGSprecip is also calculated from δ2Hhair; Smith, 2009) and because both variables contain measurement uncertainty (McArdle, 1988). We examined model residuals across collection dates to check for non-uniform variance (e.g., heteroscedasticity) across the period of molt. In light of the potential for delayed molt in reproductive female bats (Fraser et al., 2012), we also determined the relationship between δ2Hhair and δ2HGSprecip for female red bats collected between July 1 and August 31 and between July 1 and August 7. We performed all statistical analyses in R (R Core Team, 2013).

Results

We obtained a total of 112 red bat specimens (41 male, 67 female, 4 sex unknown) for evaluation of the relationship between δ2Hhair and δ2HGSprecip (Table S1). For male red bats, the standard deviation for δ2Hhair–δ2HGSprecip values for days 160–164 was 33.0 (Fig. 2A). The standard deviation dropped to 4.9 at day 165 and remained low (range: 0.5–7.5) between days 165 and 219 (June 14–August 7). Standard deviations were generally high between days 220 and 240 (August 8–August 28), averaging 19.9 during this period. The lower standard deviations of δ2Hhair–δ2HGSprecip values between days 165 and 219 (June 14–August 7) suggest that this is the approximate period during which male red bats are typically resident on their summering grounds and synthesize new annual pelage. Males collected before June 14 or after August 7 were more likely to have molted at a location other than where they were collected. There was no clear trend in temporal variation of the standard deviation of δ2Hhair–δ2HGSprecip values for female red bats (Fig. 2B).

Figure 2 Differences between δ2Hhair and δ2HGSprecip.

Standard deviations for (A) male and (B) female specimens of L. borealis as a function of Julian date. Dates were grouped in 5 day intervals. Solid vertical lines delineate the lowest period of variability (i.e., the estimated period of molt) for males (i.e., Julian days 165–219 or June 14–August 7).

A total of 64 male and female specimens were collected between June 14 and August 7. δ2Hhair values for male red bats exhibited a strong positive relationship with δ2HGSprecip during this period (R2 = 0.69, p < 0.001, n = 20; Fig. 3A). Assuming an identical period of molt for female red bats, δ2Hhair from females also exhibited a positive relationship with δ2HGSprecip, although the variance explained was lower than in males (R2 = 0.29, p < 0.001, n = 44; Fig. 3A). The mean slope and intercept for males (1.48 and 13.95, respectively) fall within the 95% confidence interval of the slope and intercept for females (1.29–2.21 and 5.09–30.95, respectively), and the mean slope and intercept for females (1.75 and 18.02, respectively) fall within the 95% confidence interval of the slope and intercept for males (1.07–1.89 and 1.89–26.0, respectively). For female red bats, the relationships between δ2Hhair and δ2HGSprecip for individuals collected between July 1 and August 31 (R2 = 0.33, p < 0.001, n = 46) and between July 1 and August 7 (R2 = 0.39, p < 0.001, n = 30) were stronger than the relationship between δ2Hhair and δ2HGSprecip between June 14 and August 7. When male and female bats (from June 14 to August 7) were combined, there was a strong positive relationship between δ2Hhair and δ2HGSprecip (R2 = 0.37, p < 0.001, n = 64; Fig. 3A), with no consistent trend in model variance across day of collection (Fig. 3B). Conversion of δ2Hhair values obtained from the four red bats of unknown sex (which were collected between June 14 and August 7; Table 1) to δ2HGSprecip using the combined relationship for males and females (Fig. 3A) produced δ2HGSprecip values within 5‰ of those calculated for these sites at http://waterisotopes.org.

Figure 3 Relationships of δ2Hhair and δ2HGSprecip during the estimated period of molt for males and females of L. borealis.

The relationship of δ2Hhair and δ2HGSprecip during the estimated period of molt for male (diamonds) and female (circles) red bats (A) and the resulting model residuals relative to sample collection date (B). The solid line in (A) represents the regression line for both sexes combined.

To assess the species-specific nature of the relationship between δ2Hhair and δ2HGSprecip we compared likelihood-of-origin maps produced based on the separate regression equations estimated for red bats and their sister taxa, hoary bats. For this exercise, we used as an example a representative δ2Hhair value of −40‰. For red bats we converted this δ2Hhair value to δ2HGSprecip using the relationship between δ2Hhair and δ2HGSprecip for our combined male and female sample (Fig. 3A), which yielded a δ2HGSprecip value of −34.1‰. For hoary bats, there currently exist two published estimates of the relationship between δ2Hhair and δ2HGSprecip during their presumed molting period (20 June–23 August). The first, from Cryan et al. (2004), is based on δ2Hhair data from museum specimens and estimates of δ2HGSprecip from Meehan, Giermakowski & Cryan (2004). The second, from Cryan, Stricker & Wunder (2014), contains the δ2Hhair data from Cryan et al. (2004), along with additional samples (Table 1). In Cryan, Stricker & Wunder (2014), the δ2Hhair data were recalibrated to different standards and estimates of δ2HGSprecip were derived from the same model (Bowen, Wassenaar & Hobson, 2005) that we used for deriving δ2HGSprecip values for the locations from which our red bat samples were collected. Conversion of a δ2Hhair value of −40‰ using Cryan et al. (2004) and Cryan, Stricker & Wunder (2014) yields δ2HGSprecip values of −19.2 and 3.6‰, respectively. Based on these conversions of δ2Hhair to δ2HGSprecip values, we produced likelihood-of-origin maps for each bat using the Isoscapes Modeling, Analysis, and Prediction tool, IsoMAP (http://www.isomap.org; Bowen et al., 2014). The likelihood-of-origin maps based on a common δ2Hhair value of −40‰ were substantially different when using the species-specific equations for red and hoary bats and applying a liberal estimate of uncertainty in the δ2HGSprecip values (10‰, based on the variation in our regression equations). The difference between the maps for red and hoary bats remained regardless of whether the Cryan et al. (2004) or Cryan, Stricker & Wunder (2014) relationship was used for hoary bats (Fig. 4).

Figure 4 Likelihood-of-origin maps for a δ2Hhair value of −40‰ that was transformed into δ2HGSprecip for L. borealis and L. cinereus.

The likelihood-of-origin maps (A, L. borealis and B and C, L. cinereus) were created using the geostatistical tool IsoMAP. Inset values represent the δ2HGSprecip values after transformation (using the combined relationship in Fig. 3A of this study for L. borealis (A), Cryan et al., 2004 (B) and Cryan, Stricker & Wunder, 2014, (C) L. cinereus).

Discussion

Stable isotope analysis has emerged as an important tool for studies of movement, migration, population connectivity, and habitat use of animals not amenable to traditional tracking methods (Hobson, 1999; Cryan et al., 2004; Rubenstein & Hobson, 2004; Fraser et al., 2012). However, applying isotope data to make such inferences requires accurate knowledge of the relationship between δ2Hhair and δ2H of precipitation. This relationship is often species-specific for different animals (Hobson et al., 2012), including bats (Britzke et al., 2009; Table 1), which makes it important to establish this relationship for focal species of interest or conservation concern. Given the recent impact of wind turbines on the migratory red bat, and the growing interest among conservation biologists and natural resource managers in applying stable isotopes to track the origins of Lasiurus spp. killed at wind-turbine facilities, it is essential to establish the reliability of δ2H for tracking the summering grounds of red bats. Our data showed positive relationships between δ2Hhair and δ2HGSprecip for both male and female red bats, which indicates that δ2HGSprecip values deduced from δ2Hhair may be used to infer the summering locations of bats captured (or killed) at distant sites, such as at wind turbines or on their overwintering grounds.

We estimated a period of molt of June 14–August 7 for male red bats based on comparison of δ2Hhair and δ2HGSprecip values. Greater variation in δ2Hhair–δ2HGSprecip values for male red bats collected before June 14 and after August 7 suggests that individuals collected outside of the approximate timeframe of June 14–August 7 were less likely to have molted at the site of capture. This estimated period of molt is similar to the δ2H-inferred period of molt (June 20–August 23) reported for the hoary bat (Cryan et al., 2004), a close relative of the red bat (Roehrs, Lack & Van Den Bussche, 2010). In contrast to males, there was no distinct period of low variability in δ2Hhair–δ2HGSprecip values for female red bats. This lack of a period of low variability may indicate that females molt outside of June–August, such as during migration. Another explanation is that female red bats undertake long-distance dispersal or even begin to migrate soon after molt, which would decrease our ability to detect a distinct molt period with δ2H, particularly if there exists geographic variation in the seasonal timing of molt and/or migration. Indeed, studies suggest that some female bats (including hoary bats, Cryan et al., 2004) delay molt until after parturition and lactation (Quay, 1970; Jones & Genoways, 1967) when they then synthesize pelage rapidly at the end of the growing season, within ∼2 weeks of autumn migration (Cryan et al., 2004). Regardless of its precise cause(s), the lack of a distinct period of low variability in δ2Hhair–δ2HGSprecip values for female red bats does preclude the use of δ2Hhair for identifying their summering grounds.

Within the estimated period of molt, we found significant positive relationships between δ2Hhair and δ2HGSprecip for red bats that were similar for males and females. However, the relationship between δ2Hhair and δ2HGSprecip for female red bats explained less of the variance (e.g., lower R2) compared to male red bats. The weaker relationship for females might be a function of delayed molt in reproductive females, as discussed above. Indeed, δ2HGSprecip had a stronger relationship with δ2Hhair for female red bats collected only in July and August than for females from June 14 to August 7. Although the precise timing of molt of female red bats warrants further study, the regression slopes and intercepts for males and females were not different (Fig. 3A) and there was only a small (5‰) maximum difference in δ2HGSprecip between the respective equations for males and females for δ2Hhair values ranging between −10 and −60‰. Thus, our results suggest that male red bats do not display aberrant migratory patterns or isotopic discrimination relative to female red bats (as suggested by Britzke et al., 2009) or other bat species (Table 1). These results also suggest that a single relationship may be used for conversion between δ2Hhair and δ2HGSprecip for both sexes of red bats. A single relationship applicable to either sex implies that this approach may be used for assessing the origin of red bats of unknown sex. For example, δ2HGSprecip values derived from δ2Hhair values for four red bats of unknown sex in our dataset (Table 1) were within 5‰ of the actual δ2HGSprecip values at these sites, which is less than the estimated uncertainty (10‰) in the relationship between δ2Hhair and δ2HGSprecip for red bats.

In contrast to our results, Britzke et al. (2009) found a negative relationship between δ2Hhair and δ2HGSprecip for male red bats. Although the precise reason for this discrepancy is uncertain, we offer two potential explanations. The Britzke et al. (2009) dataset included samples from red bats collected between May 15 and August 1 during the years 2001–2005, whereas we identified a molt period of June 14–August 7 using samples from the years 1900–1972. Thus, one explanation for these differing results is that some of red bats analyzed in Britzke et al. (2009) may have been sampled before they reached their summering grounds and molted new pelage, which means that δ2Hhair values from such bats would partly indicate their location the prior summer rather than of the year in which they were collected. A second possible explanation is that bats used in Britzke et al. (2009) were sampled across a smaller number of years. Although there is no long-term trend in δ2HGSprecip during the last ∼100 years (Hobson et al., 2010; Hobson et al., 2014), there can be inter-annual spatial variation in δ2HGSprecip. Such variation may be minimized when using samples from a large number of years (i.e., 1900–1972), whereas it may have a larger impact when using samples from a relatively small number of years (i.e., 2001–2005).

Our results provide confidence for using δ2Hhair to identify the location of the summering grounds (i.e., the location where new pelage was synthesized) of red bats of unknown geographic origin. In contrast to intraspecific similarities, our results underscore the species specificity of the δ2Hhair and δ2HGSprecip relationship, even among closely related bat species. For example, a δ2Hhair value of −40‰ yielded distinct δ2HGSprecip values and likelihood-of-origin maps for red and hoary bats based on using the regression presented here for red bats and those of Cryan et al. (2004) and Cryan, Stricker & Wunder (2014) for hoary bats (Fig. 4). Thus, our study provides critical calibration data for the use of δ2Hhair to infer the movement and migration patterns of red bats, and will enable future studies on red bat ecology and conservation, especially in the context of assessing the impacts of threats associated with global environmental change.

Supplemental Information

Table S1 Data for L. borealis museum specimens. δ2H values are presented in ‰ notation relative to VSMOW-SLAP

Click here for additional data file.

We thank John Hoogland for providing feedback on an earlier version of the manuscript. Suzanne Peurach at the Smithsonian National Museum of Natural History facilitated the sampling of museum specimens and Robin Paulman assisted with stable isotope analyses.

Additional Information and Declarations

Competing Interests

Author Contributions

Animal Ethics

The authors declare there are no competing interests.

Cortney L. Pylant conceived and designed the experiments, performed the experiments, analyzed the data, contributed reagents/materials/analysis tools, wrote the paper, prepared figures and/or tables, reviewed drafts of the paper.

David M. Nelson conceived and designed the experiments, performed the experiments, analyzed the data, contributed reagents/materials/analysis tools, wrote the paper, reviewed drafts of the paper.

Stephen R. Keller conceived and designed the experiments, analyzed the data, contributed reagents/materials/analysis tools, wrote the paper, reviewed drafts of the paper.

The following information was supplied relating to ethical approvals (i.e., approving body and any reference numbers):

The hair samples used in this research came from red bat specimens housed in the Smithsonian Institution National Museum of Natural History’s Division of Mammals Collection.

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
