# Peer review of "Stable hydrogen isotopes record the summering grounds of eastern red bats (Lasiurus borealis)"

_PeerJ, doi:10.7717/peerj.629_

## Round 0.1 · original submission · Minor Revisions

The article is well written and report interesting findings that will help red bats Conservation. Authors will have to consider some reviewers suggestion before the final acceptation of the paper.
1- It is necessary to include a section in the discussion explaining why you consider your findings are so different from Britzke et al. (2009),see Rev 1. and Rev 2. comments.
2- Compare your results with Cryan et al 2014 Ecol Appl 24:602 as it was suggested by Dr. Michael Wunder
Minor comments are included in reviewer reportings

Reviewer 1 ·

Basic reporting

This paper is generally well written. The introduction is sufficient and relevant literature is cited.

Reporting sample sizes in Table 1 would be helpful to readers.

The authors need to report estimates of the error associated with using Google Earth to determine geographic locations and elevations.

The caption for Figure 1 does not indicate what open symbols represent.

Include Figure S1 in the main manuscript, not as supporting.

Experimental design

Acceptable.

Validity of the findings

The authors need to specifically and directly address why their findings are so different from Britzke et al. 2009.

Lines 193-194 and 207-208 are not supported by the data. That is, the relationship for females is weak. Such conclusions overstate what the results actually support, especially in light of previous studies.

The authors also need to discuss the confounding influence of age and body condition.

·

Basic reporting

The article is well written and provides sufficient background and appropriate context for understanding the importance of the study. The structure is appropriate, and figures were appropriate and well described (although Figure 1 would benefit from the use of color). A broader comment:
1. As the authors point out, quantifying the relationship between isotopic ratios of hydrogen in hair and precipitation for eastern red bats is critical to understanding migratory connectivity in light of fatalities at wind power facilities. This work follows on that of Britzke et al. (2009), who also quantified the same relationship in the same species (along with others). This ms uses the work of Britzke et al. to set up the objectives, but then ignores the differences among studies entirely in the Discussion. Given that Britzke et al. found a different pattern, how do the authors of this study reconcile the divergent conclusions about the strength and direction of the relationship between hair and precipitation values for male eastern red bats? Although not stated, this ms utilizes hair samples from museum specimens that were presumably collected over a wide range of years (it would be helpful to have the specimens and relevant collection data listed in a supplementary table), while Britzke et al. used hair samples collected over a much shorter time scale. Given that isotopic ratios in precipitation may vary temporally and spatially, could the differences among studies be explained by different sampling (given that neither study has great sample sizes)? or could the difference be based on within-year date ranges of sampling (the date range was wider in Britzke et al. 2009)? Could these authors re-analyze data from Britzke et al. from the same date range to see if the pattern might be congruent? If not either of these options, do the authors have other plausible explanations? or at least ideas as to future work that could resolve the disparities among studies?

Experimental design

A few comments:
1. . While the pattern of lower SD between June 14 and August 7 in Figure 2A is evident, I would suggest using statistics that allow one to assess whether differences in variability are real (rather than simply eyeballing numbers that are based on small sample sizes). Further, low variability might suggest that molt is occurring during that time period, but only if the deltaH-hair values are close to the deltaH-precip values for the same location (one could have low variability but large differences between mean values of hair and precipitation). Could the authors use an approach like that of Cryan et al. (2004) to assess the timing of molt?
2. L117 - please explain what is meant by symmetry
3. I was a little confused in L150-161, as the wording suggests that the authors are predicting deltaH-precip from deltaH-hair, which isn't quite correct (the independent variable in Figure 3A is deltaH-precip). To assign animals to their origin we do indeed measure deltaH-hair and then based on the regression equation calculate the deltaH-precip for the likely region of origin. It would be helpful to have the language clarified. The language in L154-156 is also a bit unclear (i.e., '...we produced likelihood-of-origin maps for these deltaH-precip values...' doesn't explain that the authors took the deltaH-hair values for their specimens and calculated deltaH precip values using the two different regression equations). Figure S1 shows that the use of different regression equations results in different likelihood surfaces and hence the need to use species-specific equations, but it should be pointed out that the 'hottest' parts of the likelihood surface for the L. borealis regression equation don't exactly overlap the spatial distribution of samples used in the analysis (as in Figure 1), particularly in the south, highlighting limitations of this approach.

Validity of the findings

No major comments beyond 1) needing to be clear about the timescale of sampling and how that might influence the observed relationships between precipitation and hair values; and 2) needing to place the findings in this study in the context of previous work on the same species.

Additional comments

Just a few minor suggestions for the Introduction:
1. L11-12: there are many bat species that roost in trees that are not long-distance migrants (e.g., E. fuscus, M. lucifugus, M. septentrionalis, M. evotis, M. volans, etc.). Further the use of 'tree bats' is a terrible term, as it is normally applied to foliage-roosting species such as members of the genus Lasiurus, but ignores the fact that many other species utilize trees (and hence should also be called 'tree bats'). I would suggest simply discussing long-distant migratory bat species without invoking this useless term.
2. L48: Britzke et al. 2009 also provide data on Myotis lucifugus, not just Sullivan et al. 2012 (as in Table 1). The following sentence (L49-L51) requires citations.
3. L52-L54: Are L. borealis and L. cinereus the two main species affected by wind energy? What about high levels of mortality of silver-haired and tricolor bats (and why make the distinction)? I would suggest just stating that eastern red bat are among the most highly-affected by wind power.
4. L54-L57: Cryan (2003) should be cited at the end of this sentence.

·

Basic reporting

The writing is clear and concise. There are two points of clarification that are required. First, the term fractionation is inappropriately used. "Discrimination" is a better term for the process that the authors reference. Second, the authors should clarify what sort of in house keratin standard they used by providing the name, the material type (organic or inorganic), and the accepted delta value.

Experimental design

I had no concerns about the experimental design.

Validity of the findings

The only concern I had about the validity of the finding relates to a comparison of their calibration model to that presented by Cryan et al. (2004). The 2004 paper used a different model for the hydrogen values in precipitation. There is a newer paper published in Ecological Applications this year (Cryan et al 2014 Ecol Appl 24:602) that reports a model relating hydrogen values in hoary bat fur to the modeled values from the same Bowen model that the authors used. The authors should compare their results with the model of Cryan et al (2014) rather than Cryan et al (2004).

Additional comments

Review of Pylant et al. “Stable hydrogen isotopes record the summering ground of eastern red bats (Lasiurus borealis).

This short well-written descriptive paper relates values of stable hydrogen isotopes in bat fur to modeled values for stable hydrogen isotopes in precipitation. The data are a welcome contribution for future reference, as are the estimated relationships between the bat fur values and those predicted for precipitation. I have only a few minor technical comments that will hopefully improve the usefulness of the work.

Isotopic fractionation refers to a very specific process that is not possible to quantify by the methods in this paper. The more appropriate term to describe the difference between rain hydrogen isotopes and fur hydrogen isotopes is “discrimination”, which is much more general. I would suggest replacing fractionation with discrimination throughout the manuscript.

Please report the name and material that was used for the in-house keratin standard. Was there only one and was it organic? The USGS standards are not organic and therefore less appropriate for use in the calibration of organic standards. Typically, two different organic keratins are used for calibration, and they are reported by name, material, and also value. This is helpful because it lets researchers determine the extent of extrapolation that was necessary to predict the values of the bat fur, all of which were fairly enriched in 2H. For example, was your organic standard in the -10 to -20 per mil range, or was it down near -100?

Finally, the regression equation used in Cryan et al (2004) for hoary bats are not comparable to that presented here. It is true that the relationship is for a different species, and the authors’ point is clear and convincing that it doesn’t work to use a calibration for hoary bats to predict origins of red bats. But that’s not what was compared here; Cryan et al (2004) uses a different precipitation model than the Bowen model used by the authors. Cryan et al (2014; Ecological Applications 24: 602-14) presents a revised relationship that relates hoary bat fur to the Bowen model for precipitation. Please use that one for comparison, as it would be a better and more direct comparison.

I hope these comments help.
Mike Wunder, Denver, CO

---

## Round 0.2 · accepted · Accept

Your paper is ready to be published, you already have included the main suggested changes by reviewers